# Baroreflex Sensitivity Assessment Using the Sequence Method with Delayed Signals in End-Stage Renal Disease Patients

**DOI:** 10.3390/s23010260

**Published:** 2022-12-27

**Authors:** Marisol Martinez-Alanis, Martín Calderón-Juárez, Paola Martínez-García, Gertrudis Hortensia González Gómez, Oscar Infante, Héctor Pérez-Grovas, Claudia Lerma

**Affiliations:** 1School of Engineering, Universidad Anáhuac Mexico, Huixquilucan 52786, Mexico; 2Plan de Estudios Combinados en Medicina, Faculty of Medicine, Universidad Nacional Autónoma de México, Mexico City 04510, Mexico; 3Department of Electromechanical Instrumentation, Instituto Nacional de Cardiología Ignacio Chávez, Mexico City 04480, Mexico; 4Servicio de Radioterapia y Física Médica, Instituto Nacional de Ciencias Médicas y Nutrición Salvador Zubirán, Mexico City 04480, Mexico; 5Department of Physics, Faculty of Sciences, Universidad Nacional Autónoma de México, Mexico City 04510, Mexico; 6Department of Nephrology, Instituto Nacional de Cardiología Ignacio Chávez, Mexico City 04510, Mexico

**Keywords:** baroreflex sensitivity, estimation error, end-stage renal disease, hemodialysis, heart rate

## Abstract

Impaired baroreflex sensitivity (BRS) is partially responsible for erratic blood pressure fluctuations in End-Stage Renal Disease (ESRD) patients on chronic hemodialysis (HD), which is related to autonomic nervous dysfunction. The sequence method with delayed signals allows for the measurement of BRS in a non-invasive fashion and the investigation of alterations in this physiological feedback system that maintains BP within healthy limits. Our objective was to evaluate the modified delayed signals in the sequence method for BRS assessment in ESRD patients without pharmacological antihypertensive treatment and compare them with those of healthy subjects. We recruited 22 healthy volunteers and 18 patients with ESRD. We recorded continuous BP to obtain a 15-min time series of systolic blood pressure and interbeat intervals during the supine position (SP) and active standing (AS) position. The time series with delays from 0 to 5 heartbeats were used to calculate the BRS, number of data points, number of sequences, and estimation error. The BRS from the ESRD patients was smaller than in healthy subjects (*p* < 0.05). The BRS estimation with the delayed sequences also increased the number of data points and sequences and decreased the estimation error compared to the original time series. The modified sequence method with delayed signals may be useful for the measurement of baroreflex sensitivity in ESRD patients with a shorter recording time and maintaining an estimation error below 0.01 in both the supine and active standing positions. With this framework, it was corroborated that baroreflex sensitivity in ESRD is decreased when compared with healthy subjects.

## 1. Introduction

The leading cause of mortality in end-stage renal disease (ESRD) is cardiovascular disease [1,2]. The pathophysiological changes that contribute to cardiovascular events in this group of patients include hydric overload, hypertension, and increased vascular resistance [3]. ESRD is considered to be a model of the extreme hemodynamic adaptations of physiological mechanisms that are yet able to cope with the major hemodynamic challenges induced by hemodialysis (HD) [4,5]. The autonomic nervous system modulates cardiovascular function to bound blood pressure variability in the short term [6] and contributes to the hemodynamic stability of ESRD patients [7]. However, autonomic dysfunction is a common consequence of ESRD [8].

Changes in the baroreflex function, including the baroreflex sensitivity (BRS), may reflect alterations in the autonomic control of the cardiovascular system [6,9,10]. Compared that of the healthy population, the BRS in ESRD patients is significantly decreased due to many combined factors, including age, arterial stiffening, damaged peripheral pressure sensors [11], and autonomic neuropathy [12,13]. It is also known that BRS is reduced in hypertension [11], atrial fibrillation [14], and other cardiovascular affections, including diabetes, obesity, and cardiac failure [15,16]. The clinical value of BRS has been demonstrated, since it is an independent mortality risk factor in ESRD patients [17].

After recognizing that baroreceptors are stimulated not only by abrupt pressure changes but by small, continuous fluctuations during daily life, low-cost methods that can be used in a noninvasive fashion to analyze the spontaneous beat-to-beat variability have been pursued and proposed. The BRS index can be estimated by extracting it from the power spectral analysis [9,11,16] or simply using the sequence method in interbeat interval (IBI) time series, calculating the mean slope of all sequences for either the positive or negative sequences [9]. The active orthostatic test is a noninvasive maneuver allowing for an extra assessment of BRS, as it provokes a sympathetic reflex that increases heart rate, arterial resistance, and blood pressure [18,19]. The impairment of baroreflex causes a decreased ability to alter the cardiac period in response to acute alterations in blood pressure and a reduced ability to buffer changes in systemic blood pressure. This experimental design lets us evaluate BRS changes that follow a physiological adjustment, like the active orthostatism in two controlled conditions: healthy and chronically ill patients [20]. The diminishment of BRS in ESRD patients has been previously documented, however, this population is usually under the influence of pharmacological antihypertensive treatment, which may have an influence on instantaneous BRS measurements [12,21,22,23].

We propose to use the delayed sequence method where the studied time series allows for the estimation of BRS with an optimal recording time shift, employing shorter time series than the 15 min traditionally required for the sequence methods in healthy subjects [24]. However, the feasibility of this method in ESRD has not been demonstrated. The purpose of this work was to assess BRS through this delayed sequence method in ESRD patients without pharmacological antihypertensive treatment during an active orthostatic test and to compare it with clinically healthy subjects.

## 2. Materials and Methods

### 2.1. Study Protocol and Participants

The study included one group of 22 healthy volunteers and consecutively recruited 18 ESRD patients treated with hemodialysis thrice per week, for at least 3 months. The description of the population is shown in Table 1. None of the participants were smokers. Healthy volunteers did not take any medication. According to standard clinical management of ESRD patients undergoing chronic HD in our centre, the patients did not take erythropoietin, any antiarrhythmics, or antihypertensive drugs since blood pressure control in our centre is achieved solely with HD prescription [5,25]. The study on ESRD patients was performed 60 min before an HD session with a preceding interdialytic period of 48 h. The etiology of primary renal disease was diabetes mellitus in only three patients (16.6%). All participants were asked to refrain from caffeine and alcohol 12 h before the study protocol. All procedures followed the ethical standards of the Research and Ethics Committee of the Instituto Nacional de Cardiología Ignacio Chávez (protocol number 12-763) and the 1964 Helsinki declaration and its later amendments. Written informed consent was obtained from all participants.

We followed the study protocol described previously [24,26]. Blood pressure was recorded noninvasively with a Finometer (Finapres Medical Systems) at 200 samples per second. The Finometer was calibrated with a compressive sphygmomanometer. The thimble was placed in the middle finger of the left hand, with the arm resting on a sling. The active orthostatic test was conducted as follows: (1) the subjects were placed in supine position 5 min before the recording of blood pressure and continued for 15 min; (2) the participants stood up by themselves, 5 min after the recording lasted 15 min in this position. 

### 2.2. Signal Processing and Beat Identification

From the digitized blood pressure recordings, the last 15 min of recording in each position (supine position and active standing) were copied and stored into separate files for further processing. The blood pressure recordings were filtered using the second derivative method [27], and they were rectified to obtain only positive values. A threshold was set out to find the peaks of the signal corresponding to the systolic blood pressure (SBP) value of each heartbeat (Figure 1, panel A). Then, for each heartbeat, the pulse wave starting point (indicated with a magenta triangle in Figure 1, panel A) and the IBI time series were obtained by subtracting the time of consecutive pulse wave starting points (Figure 1, panel A).

### 2.3. Baroreflex Sensitivity Estimation

Using the SBP and IBI signals, the BRS index was calculated using the sequence method [28]. The region of interest of the studied signals is in the low-frequency band (0.04 to 0.15 Hz), so a low-pass filter with a cutoff frequency of 0.15Hz was used to smooth out the SBP and IBI time series. This filter attenuates the influence of respiration on SBP and IBI fluctuations and privileges fluctuations in the low-frequency band that are associated with baroreflex modulation [15]. Each sequence corresponds to a group of at least four consecutive beats in either ascending or descending order for both the SBP and IBI time series (Figure 1, panel B). Dispersion plots were used to graph the sequences (IBI vs. SBP), and the least-squares method was used to calculate the linear regression of each sequence. The BRS index was then calculated as the average of all the sequences in either a positive or negative direction (Figure 1, panel C). To consider the effect of the baroreflex delay, all signals were shifted from 0 to 5 beats, as shown in [24]. Knowing that changes in SBP occurred before changes in IBI, the delay was applied to SBP according to IBI [24].

To calculate the optimal recording time, the standard error of the slope (ξ) for each sequence was calculated using Equation (1):(1)$$\xi =\sqrt{\frac{\sum_{i=1}^{N_{p}}{\left(IBI_{i}-\hat{IBI_{i}}\right)}^{2}}{\left(N_{p}-k-1\right)\sum_{i=1}^{N_{p}}{\left(SBP_{i}-\bar{SBP_{i}}\right)}^{2}}}$$
where $N_{p}$ is the total number of points, $IBI_{i}$ is the IBI value for heartbeat number i, $\hat{IBI_{i}}$ is the IBI estimated value for heartbeat i (using the linear regression), k represents the degrees of freedom (in this case, k = 2), $SBP_{i}$ is the SBP value for heartbeat i, and $\bar{SBP_{i}}$ is the average SBP value for the sequence.

The reference error ($\xi_{ref}$) was calculated for all sequences contained in the time series recording and averaged, obtaining two reference errors: one for positive sequences ($\xi_{ref+}$) and one for negative sequences ($\xi_{ref-}$). The reference signals were then split in half and the first half of the signals were used to calculate the test errors for both positive ($\xi_{test+}$) and negative sequences ($\xi_{test-}$). The test error was kept within the margin of the reference error. If the test error exceeded the reference error, the second half of the signal was cut again in half and added to the test signal in order to recalculate the error. Once the test signal was shorter than 3 min or the second half of the signal was shorter than 10 s, the optimal time for the recording was selected. Both SBP and IBI time series were adjusted to this recording time and the BRS indexes were calculated. If there were no positive or negative sequences in the new recordings, a previous optimal time was considered. A more detailed explanation of this process can be found in the algorithm in Figure 2. The optimal recording time and its corresponding BRS indices were obtained for all subjects in both supine position and active standing, as well as for the original time series and all the delayed signals. Determination coefficient is equal to 1 minus the quotient of the sum of square residuals and the total sum of squares. The number of data points is the quantity of heartbeats included in a sequence. In turn, the number of sequences refers to the set of three or more consecutive beats that increase or decrease in SBP and IBI. 

### 2.4. Heart Rate Variability

Statistical heart rate variability indices obtained from 15-min time series were calculated to provide a common analysis background. We show the mean value (MeanIBI) and standard deviation (SDIBI) of IBI values [29].

### 2.5. Statistical Analysis

Results are shown as median (percentile 25–percentile 75). Our data has abnormal distributions according to the Kolmogorov-Smirnov test. Categorical values were compared using Fisher’s exact test. Quantitative variables were compared between groups, and within positions by Wilcoxon rank sum and signed rank (for paired observations) tests. Statistical significance was considered *p* < 0.05 [30]. Statistical analysis was performed with MATLAB version 2020a (https://la.mathworks.com/products/matlab.html, accessed 1 November 2020) and SPSS version 26 (https://www.ibm.com/products/spss-statistics, accessed November 2018).

## 3. Results

Table 2 shows that the mean SBP is greater in ESRD patients than in the healthy group, as well as in the active standing compared with the supine position. It is also observed that the MeanIBI (supine position) and SDIBI (supine position and active standing) are larger in healthy subjects compared with ESRD patients. The MeanIBI and SDIBI are also larger during the supine position than during active standing (for both healthy and ESRD patients).

Figure 3 shows that the BRS values were higher in healthy subjects (black) in the supine and active standing positions compared with those of ESRD (gray) patients (all delays). The BRS of healthy subjects was higher in the supine position when compared to active standing (positive and negative sequences). Regarding the number of data points in positive sequences, it was higher in ESRD patients during the supine position compared to active standing with delays 2, 4, and 5. The BRS with negative sequences was higher with all delays in the supine position. The number of data points (heartbeats) increased during active standing in the healthy and ESRD groups (Figure 4). 

The number of sequences (Figure 5) was larger in healthy subjects than in ESRD patients in the supine position (positive sequences: delays 0–2, 4, and 5; negative sequences: 0–3). The number of sequences was also larger in the active standing compared with the supine position in both healthy subjects and ESRD patients, with most of the delays. The estimation error (Figure 6) remained below 0.009 in all cases. This value was higher in healthy subjects compared to ESRD patients, during both the supine and active standing positions. In the supine position, the estimation error was generally higher in healthy subjects and ESRD patients.

Few differences are observed between the healthy and ESRD groups regarding the determination coefficient (Figure 7). Some differences can be observed with positive and negative sequences, between the supine position and active standing in healthy subjects. The optimal recording time remained approximately the same for all delays and groups (Figure 8).

## 4. Discussion

BRS measurement using the sequence method for spontaneous SBP and IBI fluctuations is widely known [9,16,28]. The use of delayed signals has been shown to be reliable in the BRS assessment of healthy subjects and patients with different conditions, including ESRD treated with antihypertensive medication [12,31,32]. In this work, we show the assessment of BRS in ESRD patients without the influence of antihypertensive medication. This method relies on the estimation error to ensure a reliable assessment of BRS even when time series are shorter than the standard recording time of 15 min.

The modified method showed that the BRS of ESRD patients, which is already reduced during resting conditions (supine position), does not decrease significantly when facing the active orthostatic challenge when positive sequences are used, in contrast with the clear reduction of BRS in the healthy group during active standing (Figure 3). The orthostatic challenge increased the number of data points and the number of sequences in both groups (Figure 4 and Figure 5). The estimation error (Figure 6) was consistently lower in ESRD patients than in the healthy group under all conditions (i.e., with all delays and both body positions), despite the similar number of data points and sequences between groups (Figure 4 and Figure 5). This lower estimation error in the ESRD group could not be explained by a better goodness of fit of the individual sequences, since the determination coefficient (R^2^) was similar between groups under most conditions.

Regardless of the differences between the groups in the estimation error, the modified delayed sequences method allowed for the expected estimation of BRS with a time series shorter than 15 min, with a similar optimum recording time between groups and conditions (Figure 8).

The reduction of the optimal recording time, with an estimation error in delayed signals below the original data error, may shorten the time needed for BRS calculation during the active standing test, which is often uncomfortable in patients with more deteriorated health, and minimize the stress of actively standing for a long time.

BRS is an important mechanism of hemodynamic stability in patients undergoing chronic hemodialysis. It is known that peripheral vascular resistance is higher in patients prone to develop intradialytic hypotension, which also has an impaired BRS response [23]. A myriad of determinants modify the BRS in ESRD patients, such as the renin-angiotensin-aldosterone system, vascular resistance, uremic toxins, anemia, sodium and water retention, mineral metabolism, and endocrinological disruption; the combination of these factors causes cardiac damage, which eventually leads to heart failure, ischemic heart disease, and sudden death [32]. In this work, we document a diminished BRS in ESRD, as other authors did [23,33]. BRS is a clinically relevant measure of cardiovascular status, as it has been reported that it is a predictor for all-cause mortality and sudden death in the context of chronic kidney disease on hemodialysis, peritoneal dialysis, and conservative treatment [17].

As mentioned above, the pathophysiological pathways for the disruption of BRS are diverse. Additionally, with aging, cardiac and total body noradrenaline spillover and skeletal muscle sympathetic outflow increase. That is, the delivery of noradrenaline to its receptors increases [15,34]. This process could be augmented in ESRD patients, as it is reported that they have impaired autonomic adjustments characterized by the overactive sympathetic nervous system and reduced parasympathetic modulation [35]. Although none of the patients had a previous history of malignancies, head and neck malignancies are a possible cause of BRS failure that are often missed during clinical evaluation [36,37].

Baroreflex sensitivity is markedly reduced in middle-aged and juvenile patients with ESRD due to the combined effects of aging, arterial stiffening, and autonomic neuropathy [12,38]. In particular, Ye K and collaborators report that ESRD patients have orthostatic stress in a lower body negative pressure test [35] and increased cardiovascular risk [17,35,39]. These are risk factors for the development of endothelial dysfunction and peripheral arterial disease [39]. However, our group has shown that, in a particular group of ESRD patients without hypertensive or any other drug treatment, there is a preservation of autonomic regulation before active orthostatism [5]. Complementary to our previous work, nonlinear measures of heart rate variability also show restricted IBI variability in ESRD compared to healthy subjects [40]. ESRD comprises a broad group of clinical manifestations in and pathological modifications to metabolism. In this advanced stage of the syndrome, patients lose weight due to chronic inflammation, malnutrition, and protein-energy wasting [41]. Therefore, this set of patients has a lower average BMI than the healthy population, as observed in our study.

The heterogeneity of the results when using different protocols to evaluate the integrity or deterioration of the ANS in ESRD patients leads to a search for other protocols or processing methods. Regarding the BRS and the hemodynamic control, it has been proposed that, in the closed-loop spontaneous resting conditions, the sympathetic hemodynamic control could be mainly feedforward (non-baroreflex), while the feedback, baroreflex-mediated responses occur in response to active perturbations [22,42]. However, these studies were done in animals [22]. Sapoznikov et al. were able to show a predominance of the baroreflex-mediated episodes (feedback) in healthy persons and in ESRD patients under HD treatment, and of non-baroreflex (feedforward) episodes in transplanted patients [22].

The upright posture strengthens the coupling between IBI and SAP consistently, with major participation of the arterial baroreflex, while paradoxically decreasing BRS [18]. These authors conclude that the BRS decrease linked with orthostatism is a byproduct of decreased parasympathetic modulation, triggered by the arterial baroreflex in response to central hypovolemia [18]. Most studies of BRS in ESRD patients were performed only during resting (the supine position) [12,23], and one study performed a passive orthostatic challenge [21], where no difference was found between BRS measurements during the supine position and head-up tilt position. Given the baseline reduced BRS, we expected there to be no significant change during active standing in these patients, while in the healthy controls, a significant decrease was expected, as previously reported [24]. Interestingly, we observed a significant decrease in the BRS of ESRD patients in response to active standing for different delays in the positive sequences (2, 4, and 5 heartbeats) and negative sequences (0 and 3). The discrepancy in the delays where these changes were observed between the positive and negative sequences may be due to a small effect (the decrease in BRS is small in ESRD patients), and further studies with a larger sample are needed to confirm this finding.

The limited sample size of the studied population does not permit the generalization to other sets of patients, where the use of antihypertensives and diet restrictions may modify BRS. We found a significant difference in BMI between ESRD patients and healthy subjects, and three patients had diabetes mellitus (a common cause of dysautonomia), hence, a portion of the biological effect of the active standing test may be influenced by these conditions, and further research is needed to address these issues. It is plausible that some clinical variables, such as peripheral resistance and vascular calcification, influence the measurement of BRS with the presented algorithm. This potential relation remains to be explored in future studies, as well as its clinical application in the susceptibility of intradialytic hypotension, cardiovascular risk, or the response after HD. Future studies should also consider 24-h ambulatory blood pressure measurements.

We applied low-pass filtering to the SBP and IBI time series to attenuate the effect of respiration and to enhance the low-frequency fluctuations that are associated with the baroreflex modulation [15]. The influence of the high frequency band on the traditional sequence method has been documented in healthy rats [43]. However, further studies are needed to assess the implication of the low-pass filter on BRS estimation with our method. Future research should be conducted to investigate the maximal delay at which the agreement between the new BRS values and the ones obtained by the original time series is preserved; this could diminish the time required for IBI recordings. Further studies are needed to compare our method to other methods [44,45,46]. Furthermore, this method may allow for the measurement of BRS in other pathologies in patients for which active standing may be troublesome.

## 5. Conclusions

ESRD patients with hypertension controlled only with HD show a diminished BRS in the supine position and active standing position compared with healthy subjects. These patients also have a marked BRS response during active standing, as is observed in healthy subjects. These measurements can be obtained using the modified sequence method with delayed signals and maintaining an estimation error below 0.01 while requiring shorter recordings (<15 min).

## Figures and Tables

**Figure 1 sensors-23-00260-f001:**
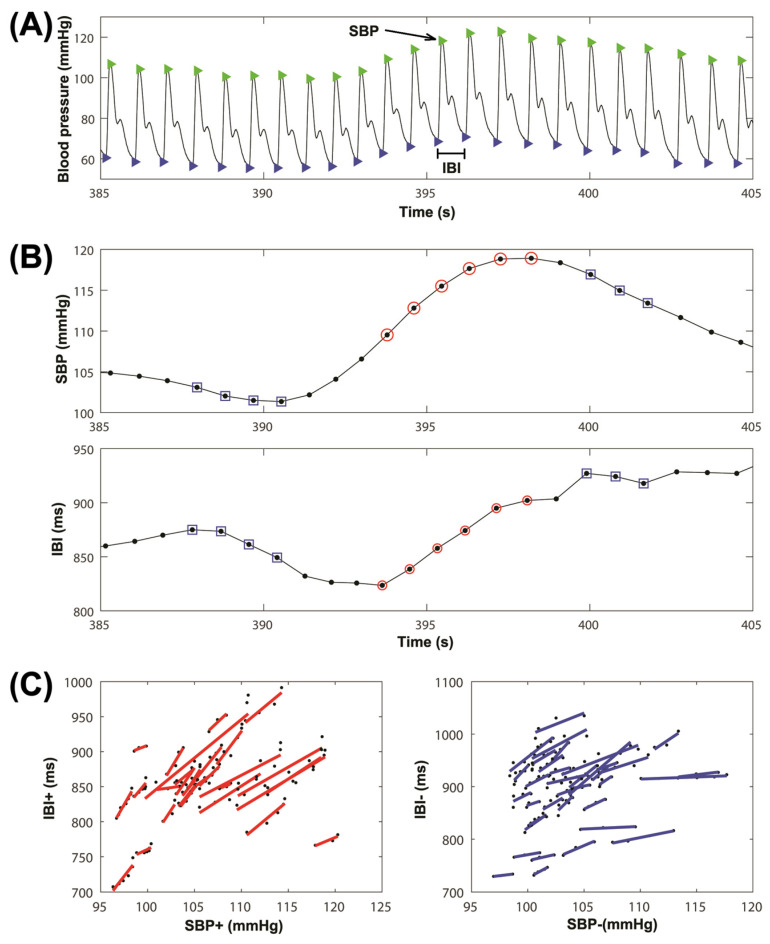
Example of blood pressure recording during supine position. (**A**) The beat identification and measurement of systolic blood pressure (SBP) and interbeat interval (IBI). (**B**) Identification of positive sequences (red circles) and negative sequences (blue squares), from filtered SBP and IBI signals. (**C**) Measurement of baroreflex sensitivity (BSR) from positive sequences (red lines in left panel with increasing SBP and IBI) and negative sequences (blue lines in right panel with decreasing SBP and IBI).

**Figure 2 sensors-23-00260-f002:**
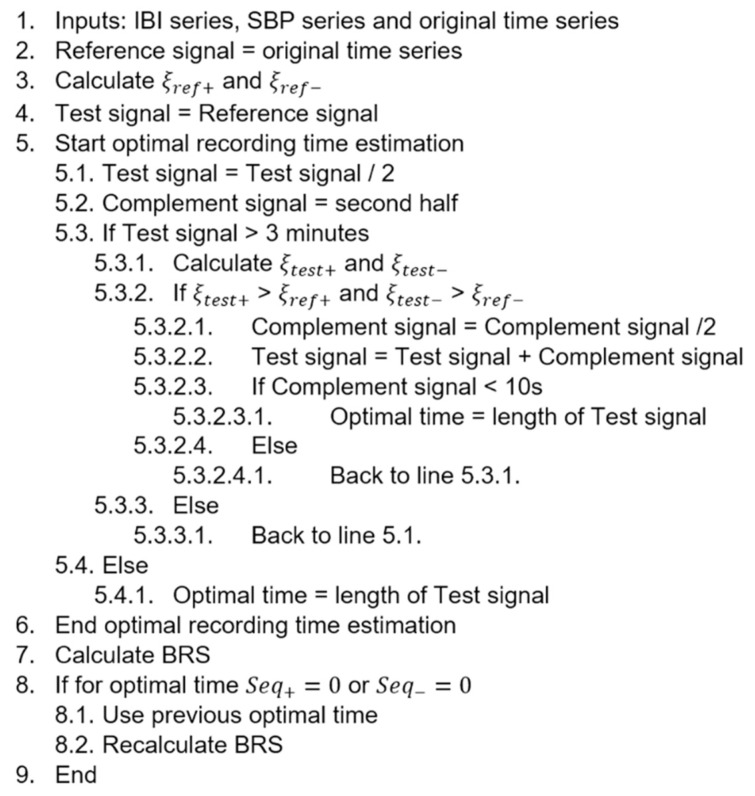
Algorithm for estimation of the optimal recording time for assessment of BRS with the delayed method.

**Figure 3 sensors-23-00260-f003:**
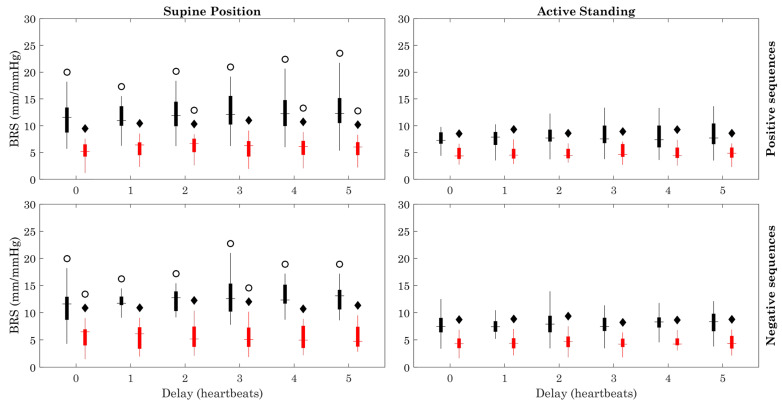
Baroreflex sensitivity (BRS). Figure symbology: Healthy subjects (black), ESRD patients (red). Black diamond: ESRD vs. Healthy group (within same position), *p* < 0.05. Open circle: Supine Position vs. Active Standing (within same group), *p* < 0.05.

**Figure 4 sensors-23-00260-f004:**
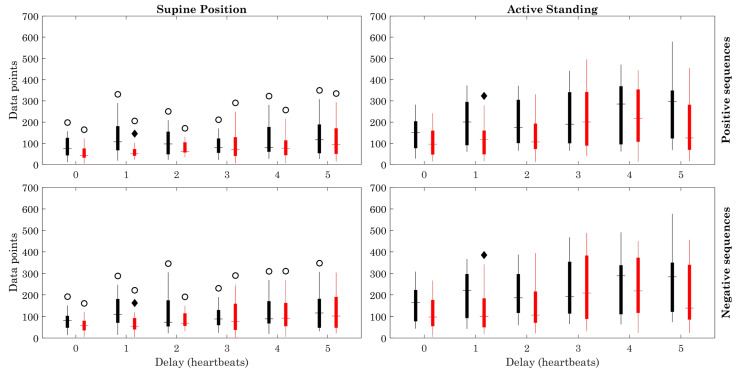
Number of data points. Figure symbology: Healthy subjects (black), ESRD patients (red). Open circle: Supine Position vs. Active Standing (within same group), *p* < 0.05. Black diamond: ESRD vs. Healthy group (within same position), *p* < 0.05.

**Figure 5 sensors-23-00260-f005:**
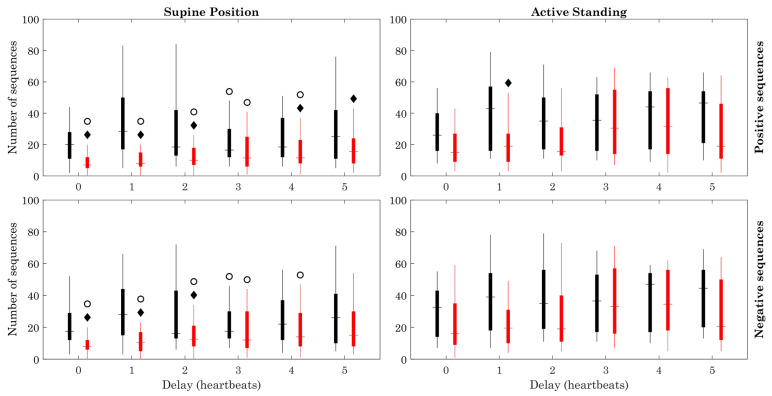
Number of sequences. Figure symbology: Healthy subjects (black), ESRD patients (red). Open circle: Supine Position vs. Active Standing (within same group), *p* < 0.05. Black diamond: ESRD vs. Healthy group (within same position), *p* < 0.05.

**Figure 6 sensors-23-00260-f006:**
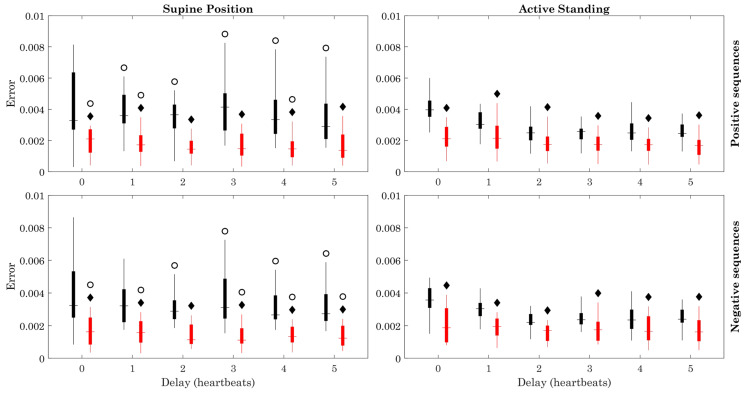
Estimation error. Figure symbology: Healthy subjects (black), ESRD patients (gray). Black diamond: ESRD vs. Healthy group (within same position), *p* < 0.05. Open circle: Supine Position vs. Active Standing (within same group), *p* < 0.05.

**Figure 7 sensors-23-00260-f007:**
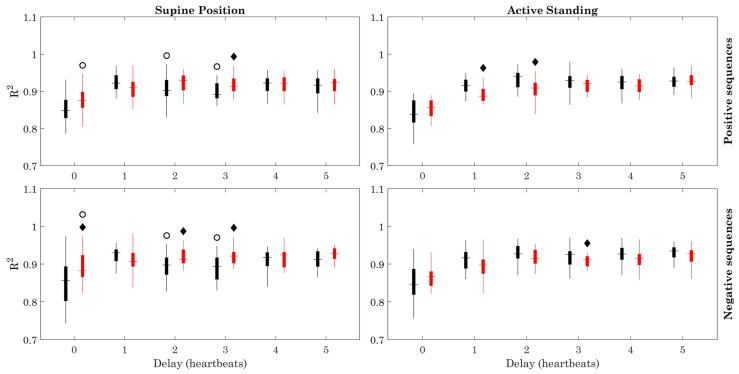
Determination coefficient (R^2^). Figure symbology: Healthy subjects (black), ESRD patients (gray). Black diamond: ESRD vs. Healthy group (within same position), *p* < 0.05. Open circle: Supine Position vs. Active Standing (within same group), *p* < 0.05.

**Figure 8 sensors-23-00260-f008:**
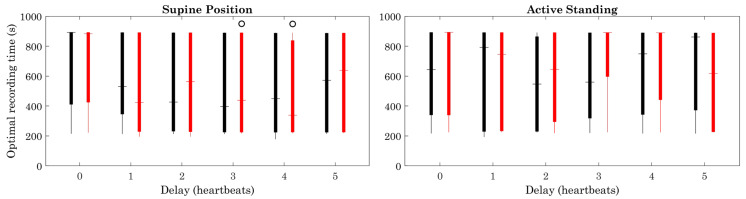
Optimal recording time. Figure symbology: Healthy subjects (black), ESRD patients (gray). Open circle: Supine Position vs. Active Standing (within same group), *p* < 0.05.

**Table 1 sensors-23-00260-t001:** Anthropometric and clinical characteristics of healthy and ESRD patients. Data shown as absolute frequency (percentage of relative frequency) or median (interquartile range).

	Healthy (N = 22)	ESRD (N = 18)	*p*-Value
Female sex	10	8	1.000
Age (years)	27 (24–33)	34 years (25–38)	0.180
BMI (kg/m^2^)	25.8 (23.7–29.2)	22.2 (19.5–26.0)	0.011
Hemodialysis vintage (months)		24 (4–24)	–
ESRD etiology			
Diabetes mellitus	–	3 (16.6%)	–
Unknown	–	15 (83.3%)	–

BMI: body mass index.

**Table 2 sensors-23-00260-t002:** Systolic blood pressure (SBP) variability and heart rate variability indices obtained from IBI time series. Data shown as median (interquartile range).

	Healthy (N = 22)	ESRD (N = 18)
	Supine Position	
Mean SBP	95.769 ^§†^ (88.696–103.571)	115.523 ^†^ (101.618–136.117)
SD SBP	5.267 (4.539–6.587)	6.580 ^†^ (4.623–8.822)
MeanIBI (ms)	943.813 ^§†^ (871.457–1012.812)	733.357 ^†^ (669.836–826.394)
SDIBI (ms)	61.097 ^§†^ (57.407–69.967)	27.493 ^†^ (16.460–38.465)
	Active Standing	
Mean SBP	109.144 ^§^ (95.261–117.528)	136.067 (114.076–148.817)
SD SBP	5.781 ^§^ (5.024–7.024)	9.851 (7.931–14.178)
MeanIBI (ms)	737.608 (719.241–767.310)	686.531 (665.392–749.893)
SDIBI (ms)	49.413 ^§^ (40.793–56.900)	28.758 (23.008–42.184)

Mean SBP is the mean value of systolic blood pressure, SD SBP is the standard deviation of systolic blood pressure, MeanIBI is the mean IBI value, and SDIBI is the standard deviation of IBI values. The ^§^
*p* < 0.05 for the Healthy vs. ESRD group (same position). The ^†^
*p* < 0.05 for the Supine Position vs. Active Standing (same group).

## Data Availability

The raw data supporting the conclusions of this article will be made available upon request to the corresponding author, provided pertinent legal requirements are met.

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
