# Peer review of "Baroreflex Sensitivity Assessment Using the Sequence Method with Delayed Signals in End-Stage Renal Disease Patients"

_sensors, 2022, doi:10.3390/s23010260_

Round 1

Reviewer 1 Report

Notes for the authors:

The study by Martínez-Alanís and collaborators investigated in healthy control volunteers and patients with end-stage renal disease the application of modified delayed signals in sequence method for BRS assessment. The study is interesting; however, I have some important considerations, mainly regarding originality.

1. What is the contribution of this manuscript to the frontier of knowledge, since in my view there is nothing new in the findings?

2. Assessment of baroreflex sensitivity by the sequence method requires a delay between blood pressure and pulse interval signals. This is not new and has already been demonstrated in different conditions. What does the study add to the current literature?

3. Based on the literature, did you expect a different result regarding impairments in baroreflex sensitivity in patients with end-stage chronic kidney disease?

4. Analysis of heart rate variability and blood pressure could enrich the manuscript.

Minor notes

1. How was the sample calculation performed?

2. The presentation of tables characterizing the studied population is important.

3. The use of black and gray fills in the graphics is almost imperceptible to discriminate between groups.

Author Response

Response to Reviewer 1

The study by Martínez-Alanís and collaborators investigated in healthy control volunteers and patients with end-stage renal disease the application of modified delayed signals in sequence method for BRS assessment. The study is interesting; however, I have some important considerations, mainly regarding originality.

Comment: 1. What is the contribution of this manuscript to the frontier of knowledge, since in my view there is nothing new in the findings?

Response: The diminishment of BRS in ESRD patients has been previously documented. However, these populations are usually under the influence of pharmacological antihypertensive treatment, which may have an influence on instantaneous BRS measurement. In this work, we employ a unique population in which hypertension is controlled solely with hemodialysis treatment, and thus, we report BRS behavior during an active standing test without antihypertensive medication. This is now mentioned in the Introduction section (lines 67 to 70) and Discussion section (lines 322 to 332).

Comment: 2. Assessment of baroreflex sensitivity by the sequence method requires a delay between blood pressure and pulse interval signals. This is not new and has already been demonstrated in different conditions. What does the study add to the current literature?

Response: As mentioned in Response 1, and now mentioned in the main text Introduction section (lines 67 to 70) and Discussion section (lines 322 to 332),  most of the studies were performed in ESRD patients with antihypertensive drugs, which may bias the assessment through their effect on the cardiac function and vascular properties. Our results offer measurements free from such bias. And we found that this particular sample of patients does have a statistically significant decrease in BRS while on active standing.

Comment: 3. Based on the literature, did you expect a different result regarding impairments in baroreflex sensitivity in patients with end-stage chronic kidney disease?

Response: Most of the BRS measurements reported previously were done in populations with mainstream pharmacological antihypertensive treatment, which may modify the BRS measurement. In our study, we measured BRS and other parameters in ESRD patients without antihypertensives, which is a unique opportunity to study BRS in ESRD. 

Many previous studies have reported decreased BRS in ESRD patients, as is shown in a Review from Kaur et al (2016) (10.5527/wjn.v5.i1.53), and therefore we expected a lower BRS in ERSD patients compared to healthy controls. Additionally, we measured the response to an active orthostatic challenge. Most studies of BRS in ESRD patients were performed only during resting (supine position) (10.1038/sj.ki.5000307; 10.1111/j.1542-4758.2009.00403.x;), and one study performed a passive orthostatic challenge (10.1371/journal.pone.0208127), where no difference was found between BRS measurements during supine position and head-up tilt. Given the baseline reduced BRS, we expected no significant change during active standing in these patients, while in healthy controls, a significant decrease was expected, as previously reported (Ref Paola). Interestingly, we observed a significant decrease in the BRS of ESRD patients in response to active standing for different delays in the positive sequences (2, 4, and 5 heartbeats) and negative sequences (0 and 3). The discrepancy in the delays where these changes were observed between positive and negative sequences may be due to a small effect (the decrease in BRS is small in ESRD patients), and further studies with a larger sample are needed to confirm this finding. This is now mentioned on page 11. 

Comment: 4. Analysis of heart rate variability and blood pressure could enrich the manuscript.

Response: We added heart rate variability and blood pressure measures in Table 2.

Minor notes

Comment: 1. How was the sample calculation performed?

Response: A sample size was not calculated a priori. However, we calculated the achieved power by GPower software version 3.1.9.7 (10.3758/bf03193146). Considering two tails testing with a normal parent distribution, alfa error probability = 0.05, total sample size = 40, and the effect size observed between BRS healthy vs ESRD (supine position, delay 0) = 0.634, the achieved power was 0.968.

Comment: 2. The presentation of tables characterizing the studied population is important.

Response: We now added Table 1 to present the characteristics of the population.

Comment: 3. The use of black and gray fills in the graphics is almost imperceptible to discriminate between groups.

Response: Gray fills were substituted for red fills to enhance contrast.

Reviewer 2 Report

The paper is overall well written and the methodology proposed here may allow an easier and faster diagnosis of impaired baroreflex sensitivity. However I have some comments and suggestions: 

- It would be interesting to perform 24h ABPM in order to test for supine hypertension, common in patients with baroreflex impairment

- I would add to the list of possible causes of baroreflex failure past or present malignancies, in particular of head and neck. Head and neck malignancies are an underecognized important cause of baroreflex failure, as shown recently, and they represent a possible cause (direct or indirect) of ESRD (PMID: 31703649). It deserves to be pointed out that baroreflex failure is an underecognized complication of head-neck malignancies +/- irradiation, with diagnostic and therapeutic implications (PMID: 31764585)

- I would better disclose the population characteristics, including medical history and home medications of ESRD patients. Drugs may interfere with the results of the Author's study, representing a possible confounder.  

Author Response

Manuscript sensors-2065482 “Baroreflex sensitivity assessment by the sequence method with delayed signals in end-stage renal disease patients”

Response to Reviewer 2

The paper is overall well written and the methodology proposed here may allow an easier and faster diagnosis of impaired baroreflex sensitivity. However I have some comments and suggestions: 

Comment: - It would be interesting to perform 24h ABPM in order to test for supine hypertension, common in patients with baroreflex impairment

Response: This is an interesting suggestion that would enrich the work, since hypertension is a major characteristic of ESRD. However, we do not have these measurements and are now addressed to our work (lines 342 to 343).

Comment: - I would add to the list of possible causes of baroreflex failure past or present malignancies, in particular of head and neck. Head and neck malignancies are an underecognized important cause of baroreflex failure, as shown recently, and they represent a possible cause (direct or indirect) of ESRD (PMID: 31703649). It deserves to be pointed out that baroreflex failure is an underecognized complication of head-neck malignancies +/- irradiation, with diagnostic and therapeutic implications (PMID: 31764585)

Response: Some causes are mentioned throughout the manuscript (lines 274 to 291), we now added head and neck malignancies as a potential cause of BRS failure (lines 291 to 293).

Comment: - I would better disclose the population characteristics, including medical history and home medications of ESRD patients. Drugs may interfere with the results of the Author's study, representing a possible confounder.  

Response: The use of antihypertensive medication is, indeed, a major feature that may modify BRS (lines 67 to 70), which are not used in this population. We now mention that patients do not take erythropoietin (line 90). We now report the known clinical history of patients in Table 1. 

Reviewer 3 Report

In this paper, authors used a method proposed in reference [20] by three of them (P.M-G., C.L., and O.I.) to assess BRS in healthy human beings. Here authors apply this method to assess BRS in ESRD patients. Some questions are raised.

1- Why authors in the abstract (as well as in the conclusions) said that sequence method with delayed signals may be useful in measurement BRS in healthy subjects, while they reported this in 2012 in ref.[20]?

2- Conclusions are a copy of the last phrase in the abstract. Please rephrase and strength it.

3- References should be updated and most recently used, as authors used 37 references, 7 of them published in the last 5 years and only 3 for the last three years. An example of recommended references is:

Ku, Tienhsiung, et al. "A Novel Method for Baroreflex Sensitivity Estimation Using Modulated Gaussian Filter." Sensors 22.12 (2022): 4618.

4- It is advised to compare the results presented in reference [22] by 5 authors of this work with the ones obtained in the present proposed method.

Author Response

Manuscript sensors-2065482 “Baroreflex sensitivity assessment by the sequence method with delayed signals in end-stage renal disease patients”

Response to Reviewer 3

In this paper, authors used a method proposed in reference [20] by three of them (P.M-G., C.L., and O.I.) to assess BRS in healthy human beings. Here authors apply this method to assess BRS in ESRD patients. Some questions are raised.

Comment 1- Why authors in the abstract (as well as in the conclusions) said that sequence method with delayed signals may be useful in measurement BRS in healthy subjects, while they reported this in 2012 in ref.[20]?

Response 1: We now changed our original statement from the abstract (line 32), as the reviewer points out, the main contribution of this work focuses on the study of ESRD and the healthy subject’s role is a control group. 

Comment 2- Conclusions are a copy of the last phrase in the abstract. Please rephrase and strength it.

Response 2: We rephrased the conclusion to summarize and explain the main findings of our work (Conclusions section, lines 357 to 362).

Comment 3- References should be updated and most recently used, as authors used 37 references, 7 of them published in the last 5 years and only 3 for the last three years. An example of recommended references is:

Ku, Tienhsiung, et al. "A Novel Method for Baroreflex Sensitivity Estimation Using Modulated Gaussian Filter." Sensors 22.12 (2022): 4618.‏

Response 3: We thank the reviewer for suggesting new and interesting literature, which is now included in our work [10].

Comment 4- It is advised to compare the results presented in reference [22] by 5 authors of this work with the ones obtained in the present proposed method.

Response 4: We now mention the complementary measures studied in our previous work [40], lines 302 to 305.

Round 2

Reviewer 2 Report

The Authors addressed all my concerns.

Reviewer 3 Report

All comments are addressed in the revised manuscript.